# PRODINFLUENCERNET: A NOVEL PRODUCT-CENTRIC INFLUENCER RECOMMENDATION FRAMEWORK BASED ON HETEROGENEOUS NETWORKS

## ABSTRACT

With the proliferation of social media, influencer marketing has emerged as a popular strategy for brands to promote their products. Recent studies have increasingly explored the use of machine learning to recommend suitable influencers for brands. This typically involves analyzing the compatibility of influencer profiles with brand attributes. However, for brands entering new markets or promoting products in unfamiliar categories, existing solutions may be limited due to insufficient information for accurate compatibility matching.

In this paper, we propose ProdInfluencerNet (PIN), a product-centric framework designed for influencer recommendation. PIN effectively models the complex relationships between brands, products, and influencers using Heterogeneous Information Networks (HINs). We categorize sponsored post images using the Google Taxonomy through image classification techniques. By leveraging the taxonomy's hierarchical structure and adopting an inductive learning approach, PIN can accurately recommend influencers for brands, even in new markets or with innovative products. We validate PIN's effectiveness and superiority over existing methods using two Instagram datasets. Furthermore, our analysis reveals that text features in profiles are more critical than images for identifying cooperative relationships between product categories and influencers.

## 1 INTRODUCTION

Influencer marketing has emerged as a dominant force in modern marketing strategies Campbell & Farrell (2020). It leverages the trust and authenticity that influencers have built with their audiences to promote products or services, often through social media platforms like Instagram.

Existing research on influencer recommendation has predominantly focused on analyzing influencer and brand profiles to identify potential matches (Gan et al., 2019; Elwood et al., 2021; Kim et al., 2023). For example, if a brand's profile expresses the brand's market and the product information they sell, and some influencer has post a number of articles related to the brand's products, then a recommendation system would likely recommend the influencer for marketing the brand's products. This profile-based matching approach may face a cold-start challenge if there is not enough information in the profiles: for example, when a brand is lunching a new product, or is entering a new market. We observe that a typical collaboration post often features the target product being promoted. By extracting the product information, we can recommend influencers for brand without resorting much to the brand's profile. In addition, by classing products into a hierarchical taxonomy, every product can be attributed to some features such that, the lower category the product can be classified into the hierarchy, the more detailed information we have about the product. As such, even when a brand is to put some new product into the market, as long as the product can be classified into the taxonomy (perhaps in a high level), we can still use the somewhat general information of the product category to find a suitable influencer to promote the product.

In this paper, we introduce ProdInfluencerNet (PIN), a product-centric framework designed for influencer recommendation. PIN effectively models the complex relationships between brands, products, and influencers using Heterogeneous Information Networks (HINs). We leverage image classification techniques to categorize sponsored post images based on the Google Taxonomy (Google, 2024).

By adopting an inductive learning approach, PIN can effectively recommend influencers for brands, even in new markets or with innovative products. We validate our framework's performance and compare it to existing benchmarks using real-world datasets collected from Instagram.

## 2 LITERATURE REVIEW

### 2.1 OVERVIEW OF INFLUENCER MARKETING

An increasing amount of research has emerged to explore various techniques to enhance the effectiveness of influencer campaigns. These studies can be categorized into several subdomains:

**Detecting undisclosed sponsorships** Detecting sponsorships in influencer marketing is crucial for maintaining transparency and trust (Villegas et al., 2023). Prior work has explored detecting undisclosed sponsorships using multimodal approaches on both Instagram (Kim et al., 2021) and X (formerly Twitter) (Villegas et al., 2023). Our experiments build upon the Instagram dataset of Kim et al. (2021), focusing on posts with confirmed sponsorships.

**Predicting post popularity** Forecasting post popularity is crucial for effective influencer marketing. Prior research has explored predicting popularity using features like image, text, and video on Instagram (Gayberi & Oguducu, 2019). Additionally, modeling user interactions on platforms like Sina Weibo has also been used to predict content popularity (Cao et al., 2020).

**Account and content classification** Analyzing influencer content and style helps brands align with suitable influencers. Previous research has classified influencers based on textual content (Nebot et al., 2018) or a combination of text and image features (Kim et al., 2020). Building on this, we leverage product-focused categorization in our work, using the cover image of commercial posts to identify the specific product being promoted.

**Influencer Recommendation** Finding the right influencers for a brand is a complex topic that goes beyond just profiling them. Some research delves into specific tiers for more in-depth analysis (Gan et al., 2019; Elwood et al., 2021; Wang et al., 2022), while others identify influencers whose target audience aligns with the brand's desired audience to maximize the impact of their campaigns (Farseev et al., 2018; Wang et al., 2022). Since influencer recommendation is central to this research, we will next provide an overview of the existing literature on this topic.

### 2.2 MACHINE LEARNING IN INFLUENCER RECOMMENDATIONS

Most research in influencer recommendation follows the general workflow illustrated in Figure 1. It begins with collecting commercial posts and account profiles from both influencers and brands. Next, features are extracted from both the post content and account profiles. These features are then fused into a multimodal representation, which is used to train a ranking model. The ultimate goal is to generate an accurate influencer ranking list for brands. While the overall process is similar, systems differ primarily in how they analyze influencer/brand features and design the model architecture.

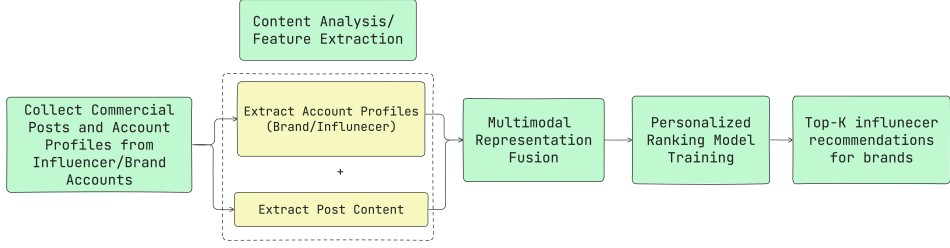

Figure 1: General Workflow for Influencer Recommendation

Researchers have explored various approaches to influencer recommendation, utilizing data from platforms like Instagram and X. For example, Farseev et al. (2018) employed psychographic user profiling, including demographics, MBTI (Schweiger, 1985), and emotion detection. However, their closed-source nature limits reproducibility. Gan et al. (2019) focused on micro-influencer ranking using multimodal embeddings and ListNet(Cao et al., 2007). Their dataset, however, is limited to micro-influencers. Elwood et al. (2021) built upon Gan et al.' work Gan et al. (2019), incorporating Tok2Vec (Honnibal et al., 2020) and VGG-16 (Simonyan & Zisserman, 2014) for feature extraction. Wang et al. (2022) proposed MORNING, a micro-influencer ranking framework incorporating target audience and cooperation preferences.

Graph-based approaches have also been employed due to their ability to capture complex relationships. Kim et al. (2023) integrated various entities into a heterogeneous network, utilizing GCN encoder (Kipf & Welling, 2016) for node embeddings. However, GCN's reliance on a fixed graph limits its applicability to dynamic social media environments. Park et al. (2024) introduced GNN-IR, employing pre-trained models for feature extraction and GraphSAGE (Hamilton et al., 2017) for link prediction. While GraphSAGE can theoretically handle unseen nodes, its real-world application remains untested. Our PIN experiment aims to address this gap and use GNN-IR as benchmark.

## 3 METHODOLOGY

### 3.1 THE SCHEMA OF PRODINFLUENCERNETWORK

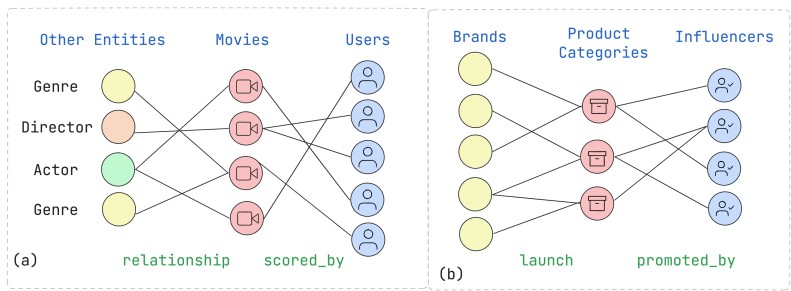

Figure 2: (a) Schema of a movie-recommendation application. (b) Our proposed network structure

Heterogeneous Information Network (HIN) is an abstraction of the real world, connecting different types of nodes through a network and emphasizing interactions between various entities (Sun & Han, 2013). Figure 2(a) illustrates a heterogeneous information network schema of movie recommendation (Yu et al., 2014), which links movies with actors, genres, and directors, as well as users that have given feedback before. Inspired by this, we propose a network structure for influencer recommendation in Figure 2(b). According to the schema, if two brands offer the same product category, they can leverage the network structure to share information about previously collaborated influencers. This expands the pool of potential influencer candidates, providing brands with a wider range of options to consider.

### 3.2 NOTATION

A heterogeneous information network $G = (V, E)$ consists of a set of nodes $V$ and a set of edges $E$ between nodes. Nodes can be further categorized into three types: influencer, product category, and brand. Given a set of brands' social media accounts $B = \{B_1, B_2, ..., B_m\}$ and influencers' accounts $K = \{K_1, K_2, ..., K_m\}$, we extract the set of products $P = \{P_1, P_2, ..., P_m\}$ launched by the brands and promoted by the influencers.

Each type of node in the network has distinct features, represented as $X_K$, $X_B$ and $X_P$ for influencers, brands, and products, respectively. The combined feature representation can be expressed as:

$$X = [X_K; X_P; X_B] \in R^{\{N \times d\}}; N = m + n + i \tag{1}$$

where $N$ is the total number of nodes of all three types, and $m, n$ and $i$ are the counts of brand, influencer, and product nodes, respectively. $d$ is the total number of node features.

### 3.2.1 OVERVIEW OF OUR ARCHITECTURE

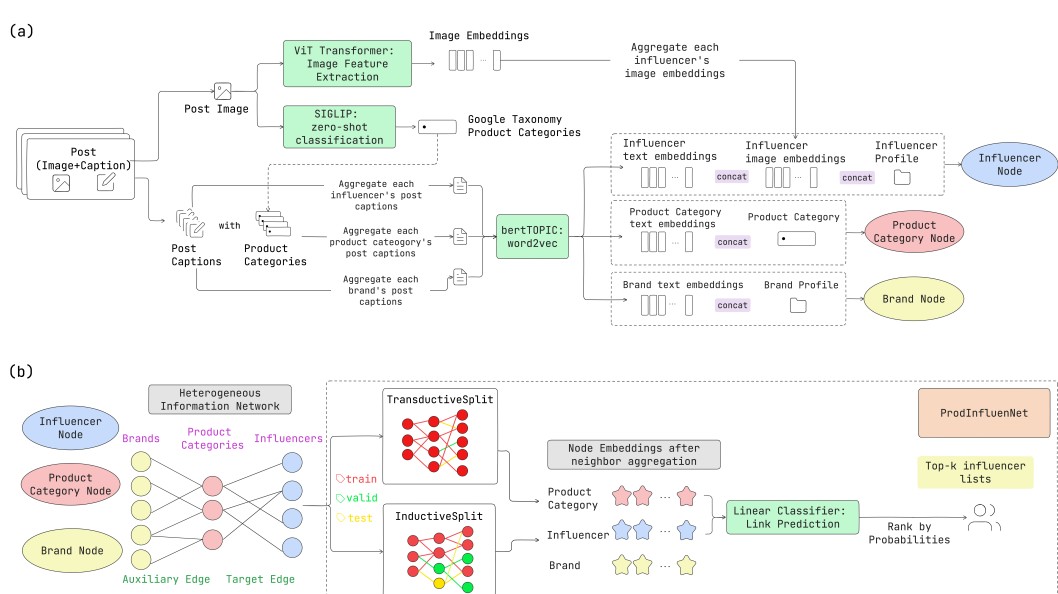

Figure 3: Our proposed framework (a) Data Pipeline (b) Graph-Based Embedding for Link Prediction and Recommendation

As shown in Figure 3, the overall structure of PIN can be divided into data pipeline and graph-based embedding for link prediction and recommendation. Since existing datasets we can obtain primarily focus on influencer and brand data, but lack information about products mentioned in collaboration posts, our data pipeline extracts features from influencers and brands and defines the product category associated with each sponsored post. With data on influencers, brands, and product categories, we can then construct the aforementioned network to facilitate link prediction tasks on the graph and subsequent influencer ranking.

### 3.2.2 GOOGLE TAXONOMY CLASS

```
Health & Beauty > Personal Care > Cosmetics > Makeup > Eye Makeup > Eye Primer
Health & Beauty > Personal Care > Cosmetics > Makeup > Eye Makeup > Eye Shadow
Health & Beauty > Personal Care > Cosmetics > Makeup > Eye Makeup > Eyebrow Enhancers
Health & Beauty > Personal Care > Cosmetics > Makeup > Eye Makeup > Eyeliner
```

Figure 4: Examples of Google Taxonomy

Before delving into our framework, we first outline our methodology for defining products within collaboration posts. We adopt Google Taxonomy, a hierarchical classification system, where products are typically categorized across 4-7 levels. Figure 4 illustrates examples of Google's taxonomy for some eye makeup products. Google Taxonomy's fine-grained detail enables us to identify specific products rather than just broader categories. For instance, within the Eye Makeup category, the last-level product categories are further classified into Eye shadow, Eye Primer, and Eyebrow Enhancers.

All products are assigned a corresponding product category from Google's product taxonomy. As previously mentioned, we also aim to address scenarios where manufacturers introduce new product types to the market. In such cases, lower-level taxonomy classes can be combined with product descriptions to form new product category nodes. By simply adding an edge linking the brand to this new node, the graph can aggregate its neighborhood and proceed with subsequent predictions.

### 3.2.3 DATA PIPELINE

Our data pipeline can further be described in three part, with two involved in using the images and captions of posts.

**Zero-shot classification of product categories**   We first classified the cover image of each post into Google Taxonomy. To avoid manually labeling product categories for each post, we opted for zero-shot classification. Specifically, we employed Sigmoid Loss for Language Image Pre-Training (SigLIP) (Zhai et al., 2023), a model that has demonstrated superior performance compared to state-of-the-art models like CLIP (Radford et al., 2021) and OpenCLIP (Ilharco et al., 2021) in both zero-shot classification and zero-shot retrieval tasks. SigLIP's advantage lies in its use of a sigmoid loss function, which operates directly on image-text pairs without requiring global normalization of pairwise similarities. This characteristic makes SigLIP a well-suited tool for our classification needs. By simply inputting the complete list of classes from Google Taxonomy, we can leverage SigLIP to obtain the most relevant taxonomy class for each post image.

**Image embedding generation for influencer features**   We utilize Vision Transformers (ViTs) (Dosovitskiy et al., 2020) as the tool of our image feature extraction task. ViTs have emerged as a powerful alternative to traditional convolutional neural networks (CNNs) for image feature extraction. Images in social media datasets are often difficult to collect due to their large size. Therefore, we aim to investigate whether incorporating image features, in addition to existing influencer metadata (followers, categories, biography, etc.), can improve prediction accuracy.

**Word2vec generation based on post caption**   We now shift our focus to converting post captions into text features. After SigLIP classification, each post is associated with a product category based on its cover image. We then aggregate all posts related to each influencer, brand, and product category, respectively, into a single document. This means that each document is composed of all the post captions associated with that entity. Utilizing BertTopic (Grootendorst, 2022), we obtain the TF-IDF representation of the document, and further convert them into text embeddings, serving as features for the respective entity. BertTopic is suited for our scenario due to its ability to discover latent topics in a corpus of documents without requiring prior knowledge or labeling. This allows us to quickly extract key points about each entity.

As illustrated in the lower right corner of Figure 3 (a), each type of nodes comprises distinct feature sets. Influencer nodes consist of text and image embeddings, as well as attributes extracted from their account profiles. Product category nodes consist of text embeddings and their corresponding class encoding. Finally, due to the absence of images in the brands' data, brand nodes contain only text embeddings and information from their account profiles. These node data was utilized in the subsequent graph construction process.

### 3.2.4 EXAMPLE OF THE DATA PIPELINE

We use an example to illustrate the data pipeline process. Consider Figure 5, where brands A, B, and C, respectively, have lunched various beauty products:

- Brand A: Perfume, Lipstick
- Brand B: Perfume, Blush
- Brand C: Perfume, Toner

We also have influencers X, Y, and Z. Each collaboration between a brand and influencer results in an Instagram post promoting the product.

The Instagram post images of promoted products are first classified into different product categories: perfume, lipstick, blush, or toner. Following this, individual documents are created for each brand (A, B, and C), each influencer (X, Y, and Z), and each product category. In the Figure, "perfume A" refers to the post caption of the perfume product launched by brand A. Similarly for "toner C", "blush B", and so on. By aggregating corresponding post captions, a brand's document contains all posts promoting that brand's products, an influencer's document contains posts published by the influencer, and a product category's document contains all posts classified within the category. This is illustrated in Figure 5, with documents colored according to their types.

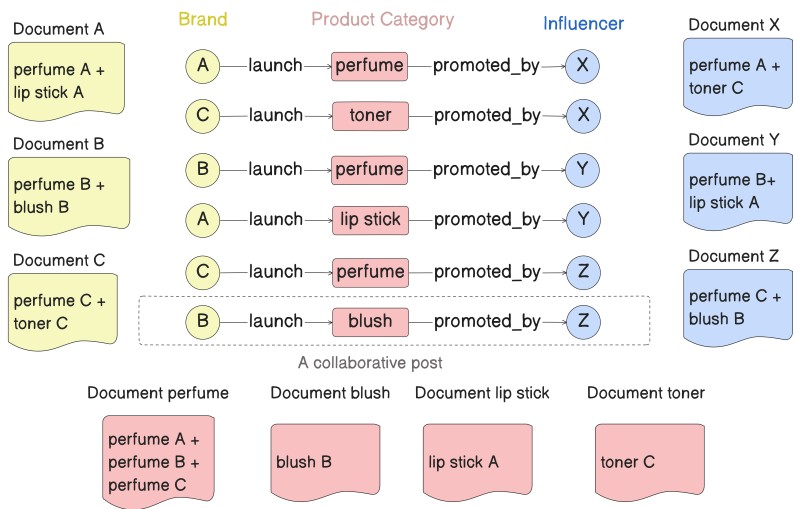

Figure 5: An example of the data pipeline

These documents were then processed using BERTopic to extract text features for each entity. These features, combined with other features specific to each node type, constituted the final graph data.

### 3.3 GRAPH-BASED EMBEDDING FOR LINK PREDICTION AND RECOMMENDATION

As illustrated in Figure 3 (b), the workflow of PIN is divided into the following two main stages:

**Graph Construction and Embedding Aggregation** The data used for graph construction can be split into two categories: data for *inductive learning* and data for *transductive learning*. In the transductive learning, all the edges connecting product categories and influencers are divided into training, validation, and testing sets. Consequently, all nodes are included in the training phase, making it unfeasible for scenarios with unseen nodes.

In contrast, in the inductive learning paradigm, we first divide the product category nodes into training, validation, and test sets based on predetermined proportions. Then, we obtain edges and their connected influencer/brand nodes linked to the product nodes. The inductive learning ensures that the model doesn't access validation or test data during training, forcing it to learn general patterns from the available features and graph structure. This approach enables the model to accommodate unseen product category nodes, thereby addressing the cold-start problem we aim to solve.

After constructing graph by different learning paradigm settings, our model utilizes a three-layer GraphSAGE Convolutional Layer, employing the Mean Aggregation Function in Eq. (2) to generate node embeddings. For a given node $v$, the function first computes the mean of the feature vectors of all neighboring nodes $u$ within the neighborhood $\mathcal{N}(v)$, represented as $h_u^{k-1}$. The vector $h_u^{k-1}$ is the representation of neighbor node $u$ at the previous layer $k-1$, which encapsulates information aggregated from its own neighborhood in the previous layer. This mean is then combined with the feature vector of node $v$ from the previous layer, $h_v^{k-1}$, ensuring that the node's own features contribute to its updated embedding. The aggregated vector, containing information from both the node $v$ and its neighbors, is subsequently multiplied by a learnable weight matrix $\mathbf{W}$ to capture important feature interactions and is passed through a non-linear activation function $\sigma(\cdot)$ to introduce non-linearity. This iterative process across multiple layers refines the node embeddings, allowing the model to encode both the local structure and higher-order information from the graph into the final node representation $h_v^k$.

$$\mathbf{h}_v^k \leftarrow \sigma\left(\mathbf{W} \cdot \mathrm{MEAN}\left(\left\{\mathbf{h}_v^{k-1}\right\} \cup \left\{\mathbf{h}_u^{k-1} \mid \forall u \in \mathcal{N}(v)\right\}\right)\right). \tag{2}$$

**Link Prediction and Ranking Probabilties**  Since our goal is to predict the existence of an edge between product category and influencer nodes, representing whether or not an influencer has promoted the product category, we utilized only the embeddings of these two types of nodes. We use Eq. (3) to compute the link_score by first multiplying the product and influencer feature vectors element-wise (edge_feat_product and edge_feat_influencer). The resulting edge_feat is then linearly transformed and the final score is obtained by summing the transformed features, which is subsequently mapped into the range of 0 to 1 using the sigmoid function in Eq. (4). With probability scores for each edge, we can rank influencers for each product category, ultimately obtaining the top-K influencer list.

$$
\begin{aligned}
\text{edge\_feat} &= \text{edge\_feat\_product} \times \text{edge\_feat\_influencer} \\
\text{reduced\_feat} &= \text{linear}(\text{edge\_feat}) \\
\text{link\_score} &= \text{sum}(\text{reduced\_feat})
\end{aligned}
\tag{3}
$$

$$
\sigma(x) = \frac{1}{1 + e^{-x}}; \; x : link\_score
\tag{4}
$$

## 4   EXPERIMENT

### 4.1   EXPERIMENT DATASETS

**Influencer and Brand (I&B) Dataset**  Kim et al. (2021)'s dataset includes details of 38,113 influencers, 26,910 brands, and over 1.6 million posts where influencers tagged brands on Instagram. Each entry contains an influencer's account, the name of the JSON file storing the post content, a list of image file names associated with the post, and a sponsorship label. The sponsorship label is used to identify whether a post is a commercial collaboration.

Since our study focuses on commercial collaborations, we neglect posts without sponsorship relationships. This initial filtering leaves over 180,000 posts for further analysis. To enhance the quality of model training and mitigate noise, we exclude influencers with insufficient data (fewer than ten collaborative posts) from the dataset. After data cleaning, our study utilizes a total of 3,281 influencers with 14,801 brands and their corresponding 70,417 collaborative posts for the experiment. After applying Google Taxonomy classification, the dataset encompasses 2,356 product categories, resulting in the generation of 45,792 Brand-launch-ProductCategory edges and 47,944 Influencer-promote-ProductCategory edges.

**iKala Dataset**  We partner with iKala Corp. to collect Instagram posts disclosing sponsorship using the branded content tag from July 1, 2022, to May 12, 2023. In total, 164,022 posts with 18 gigabytes of post metadata and 67 gigabytes of corresponding images were collected. Aligning with the I&B dataset, we filter out data with fewer than 10 collaborations and the corresponding influencers. This results in a final dataset of 15,214 brand nodes, 3,422 influencer nodes, 3,083 product category nodes, and 83,038 Brand-launch-ProductCategory edges, and 104,121 Influencer-promote-ProductCategory edges.

The initial features for each node type are illustrated in the bottom right of Figure 3 (a). The product category is represented using two key attributes: product category text embeddings (512 dimensions) and product category (11 dimensions). This results in a total dimensionality of 523 for each product category. Brands are represented using brand text embeddings (512 dimensions) and brand profile (4 dimensions), resulting in a total dimensionality of 516 for each brand. Influencers are characterized by a combination of three attributes: influencer text embeddings (512 dimensions), influencer profile (32 dimensions), and influencer image embeddings (640 dimensions). This leads to a total dimensionality of 1184 for each influencer representation.

### 4.2   EXPERIMENT SETTINGS AND EVALUATION METRICS

The experiment was set up in Python 3.9.19 and utillized PyTorch Geometric 2.5.3. Regarding hardware, a RTX 3090 GPU with CUDA version 12.2 was employed. The model configuration

was based on GraphSAGE as the backbone architecture, with 3 convolutional layers and a hidden channel size of 128. The optimization process used the Adam optimizer and a binary cross-entropy loss function, with a batch size of 1024.

The data was split into three parts for training, validation, and testing, with proportions of 80%, 10%, and 10%, respectively. Additionally, negative link sampling was performed at a 1:1 ratio relative to positive links.

Our experiments and evaluation were conducted in two parts as shown below. To facilitate a direct comparison with GNN-IR (Park et al., 2024), we aligned our metrics with theirs.

1. **Link Prediction**: The goal is to predict the existence of each link within the graph structure, along with the associated probability of the link's existence. We employed ROC AUC, Precision, Recall and F1-Score to assess the performance of our link prediction tasks.

2. **Recommendation**: We leveraged the link probabilities obtained from the previous part to generate recommendations. We evaluated the recommendation performance using rank-based metrics like Precision@K, Recall@K and F1-score@K.

Each experiment explored the following two settings, resulting in a total of six (2 * 3) experimental configurations:

1. **Learning Paradigm**
   - Transductive Learning: The model learns from the entire graph structure, including both labeled and unlabeled nodes.
   - Inductive Learning: The model learns from a subset of labeled nodes and generalizes to unseen nodes.

2. **Influencer Feature Set**
   - Text: Use only textual features extracted from influencer profiles and posts.
   - Image: Use only visual features derived from influencer images.
   - Multimodal: Combine both textual and visual features for a comprehensive representation of influencers.

### 4.3 EXPERIMENT RESULTS

The experiments were conducted on two distinct datasets, with the results presented in separate tables. Each table summarized the outcomes of the six experimental configurations applied to the respective dataset.

| Model | | ROC AUC | Prec. | Recall | F1 |
|---|---|---|---|---|---|
| $PIN_{text}$ | transductive | 0.8026 | 0.8029 | 0.802 | 0.8025 |
| | inductive | **0.9639** | **0.9474** | **0.9823** | **0.9645** |
| $PIN_{image}$ | transductive | 0.7975 | 0.8146 | 0.8009 | 0.7876 |
| | inductive | 0.7293 | 0.785 | 0.65 | 0.9882 |
| $PIN_{multi}$ | transductive | 0.8083 | 0.803 | 0.8171 | 0.81 |
| | inductive | 0.855 | 0.7751 | 1 | 0.8733 |
| GNN-IR (Kim et al., 2021) | | 0.8951 | 0.8709 | 0.8045 | 0.8364 |

Table 1: Link Prediction on I&B Dataset

| Model | | ROC AUC | Precision | Recall | F1-Score |
|---|---|---|---|---|---|
| $PIN_{text}$ | transductive | 0.8144 | 0.8323 | 0.7876 | 0.8093 |
| | inductive | **0.9516** | **0.9117** | **1** | **0.9583** |
| $PIN_{image}$ | transductive | 0.8051 | 0.801 | 0.812 | 0.8065 |
| | inductive | 0.8182 | 0.7373 | 0.9887 | 0.8447 |
| $PIN_{multi}$ | transductive | 0.8131 | 0.8227 | 0.7981 | 0.8103 |
| | inductive | 0.8433 | 0.7614 | 0.9999 | 0.8646 |

Table 3: Link Prediction on iKala Dataset

Table 1 and 2 demonstrate the performance comparison on I&B Dataset. For link prediction, models with inductive learning consistently outperform GNN-IR across all metrics, particularly excelling in recall performance. This suggests that PIN with inductive learning is capable of accurately predicting the suitability of a particular influencer for a given product by a brand. On the other hand, for recommendation, GNN-IR demonstrates high precision@K but suffers from low recall@K, indicating it might not recommend enough relevant influencers. In contrast, PIN achieves a balanced overall performance as reflected in its F1-score, consistently around 0.7, while GNN-IR's best performance is only around 0.2. This demonstrates that PIN maintains a high level of performance in both retrieving relevant influencers and ensuring recommendation accuracy.

| Model | | Metric | k=1 | k=2 | k=3 | k=4 | k=5 | k=6 | k=7 | k=8 | k=9 | k=10 |
|---|---|---|---|---|---|---|---|---|---|---|---|---|
| $PIN_{text}$ | transductive | P@K | 1 | 0.8479 | 0.7969 | 0.7679 | 0.7344 | 0.7295 | 0.7326 | 0.7334 | 0.7249 | 0.7128 |
| | | R@K | 0.2028 | 0.3192 | 0.4262 | 0.5266 | 0.6153 | 0.6651 | 0.7072 | 0.7481 | 0.7898 | 0.8334 |
| | | F@K | 0.3372 | 0.4638 | 0.5554 | 0.6248 | 0.6696 | 0.6958 | 0.7197 | 0.7407 | 0.756 | 0.7684 |
| | inductive | P@K | 1 | 0.8391 | 0.7739 | 0.7917 | 0.8523 | 0.9058 | 0.9292 | 0.9387 | 0.956 | 0.9642 |
| | | R@K | 0.4231 | 0.5956 | 0.6537 | 0.6647 | 0.6318 | 0.5943 | 0.5966 | 0.5915 | 0.5883 | 0.5915 |
| | | F@K | 0.5946 | 0.6967 | 0.7087 | 0.7227 | 0.7257 | 0.7177 | 0.7266 | 0.7257 | 0.7284 | 0.7332 |
| $PIN_{image}$ | transductive | P@K | 1 | 0.8624 | 0.7977 | 0.764 | 0.7385 | 0.7257 | 0.7274 | 0.7262 | 0.7122 | 0.7051 |
| | | R@K | 0.1908 | 0.3154 | 0.4294 | 0.5238 | 0.6103 | 0.6682 | 0.7175 | 0.7667 | 0.8108 | 0.8379 |
| | | F@K | 0.3205 | 0.4619 | 0.5583 | 0.6215 | 0.6683 | 0.6958 | 0.7224 | 0.7459 | 0.7583 | 0.7658 |
| | inductive | P@K | 1 | 0.761 | 0.693 | 0.7368 | 0.8101 | 0.8508 | 0.8634 | 0.89 | 0.9144 | 0.9169 |
| | | R@K | 0.3658 | 0.518 | 0.602 | 0.5833 | 0.5703 | 0.5853 | 0.6099 | 0.6094 | 0.6065 | 0.6116 |
| | | F@K | 0.5357 | 0.6164 | 0.6443 | 0.6511 | 0.6694 | 0.6935 | 0.7148 | 0.7234 | 0.7293 | 0.7338 |
| $PIN_{multi}$ | transductive | P@K | 1 | 0.8665 | 0.8075 | 0.7658 | 0.7322 | 0.7275 | 0.7329 | 0.7267 | 0.7121 | 0.697 |
| | | R@K | 0.1907 | 0.3158 | 0.4288 | 0.5251 | 0.608 | 0.6616 | 0.7106 | 0.758 | 0.8045 | 0.838 |
| | | F@K | 0.3203 | 0.4629 | 0.5601 | 0.623 | 0.6643 | 0.693 | 0.7216 | 0.742 | 0.7555 | 0.761 |
| | inductive | P@K | 1 | 0.7464 | 0.6667 | 0.6894 | 0.7704 | 0.8118 | 0.8537 | 0.8948 | 0.9094 | 0.919 |
| | | R@K | 0.4647 | 0.5653 | 0.6207 | 0.6353 | 0.5852 | 0.5689 | 0.5869 | 0.5632 | 0.5806 | 0.5918 |
| | | F@K | 0.6345 | 0.6433 | 0.6429 | 0.6612 | 0.6651 | 0.669 | 0.6956 | 0.6913 | 0.7087 | 0.72 |
| GNN-IR (Kim et al., 2021) | | P@k | **0.9956** | **0.9934** | **0.9926** | **0.9939** | **0.9946** | **0.9948** | **0.9951** | **0.9952** | **0.9951** | **0.9949** |
| | | R@K | 0.0268 | 0.0368 | 0.0471 | 0.0576 | 0.0677 | 0.0777 | 0.0876 | 0.0975 | 0.1072 | 0.1169 |
| | | F@K | 0.0522 | 0.071 | 0.0899 | 0.1089 | 0.1268 | 0.1441 | 0.161 | 0.1776 | 0.1935 | 0.2092 |

Table 2: Top-k Recommendation on I&B Dataset

| Model | | Metric | k=1 | k=2 | k=3 | k=4 | k=5 | k=6 | k=7 | k=8 | k=9 | k=10 |
|---|---|---|---|---|---|---|---|---|---|---|---|---|
| $PIN_{text}$ | transductive | P@K | 1 | 0.9 | 0.8627 | 0.8389 | 0.8275 | 0.8192 | 0.8469 | 0.8031 | 0.7934 | 0.7874 |
| | | R@K | 0.1033 | 0.1829 | 0.2341 | 0.2883 | 0.3336 | 0.3792 | 0.4218 | 0.4677 | 0.5071 | 0.5445 |
| | | F1@K | 0.1873 | 0.3034 | 0.3716 | 0.426 | 0.4737 | 0.5166 | 0.5575 | 0.5898 | 0.6208 | 0.6534 |
| | inductive | P@K | 1 | 0.8008 | 0.7338 | 0.714 | 0.7 | 0.6988 | 0.6944 | 0.6923 | 0.6879 | 0.69 |
| | | R@K | 0.2474 | 0.2879 | 0.3233 | 0.3565 | 0.3974 | 0.4286 | 0.4632 | 0.5388 | 0.5029 | 0.5699 |
| | | F1@K | **0.3967** | **0.4234** | **0.4473** | **0.4744** | **0.5182** | **0.5466** | **0.5588** | **0.6034** | **0.5914** | **0.6287** |
| $PIN_{image}$ | transductive | P@K | 1 | 0.8963 | 0.8675 | 0.8626 | 0.8552 | 0.8498 | 0.8423 | 0.8331 | 0.8204 | 0.812 |
| | | R@K | 0.1047 | 0.1675 | 0.2216 | 0.2729 | 0.3219 | 0.3706 | 0.416 | 0.4604 | 0.5029 | 0.5421 |
| | | F1@K | 0.1896 | 0.2823 | 0.353 | 0.4146 | 0.4677 | 0.5161 | 0.5569 | 0.5931 | 0.6236 | 0.6502 |
| | inductive | P@K | 1 | 0.8153 | 0.7474 | 0.7223 | 0.7095 | 0.7045 | 0.7072 | 0.7054 | 0.7021 | 0.7031 |
| | | R@K | 0.1384 | 0.2622 | 0.2971 | 0.3301 | 0.3691 | 0.4054 | 0.4431 | 0.4799 | 0.5129 | 0.5311 |
| | | F1@K | 0.2431 | 0.3968 | 0.4252 | 0.4533 | 0.4856 | 0.5146 | 0.5448 | 0.5712 | 0.5928 | 0.6051 |
| $PIN_{multi}$ | transductive | P@k | 1 | 0.9112 | 0.8794 | 0.8682 | 0.8583 | 0.8497 | 0.8429 | 0.833 | 0.8231 | 0.8163 |
| | | R@K | 0.1033 | 0.1685 | 0.2276 | 0.2814 | 0.3316 | 0.3816 | 0.4272 | 0.4738 | 0.5154 | 0.5575 |
| | | F1@K | 0.1873 | 0.2844 | 0.3616 | 0.425 | 0.4784 | 0.5267 | 0.567 | 0.604 | 0.6339 | 0.6625 |
| | inductive | P@K | 1 | 0.7463 | 0.7141 | 0.6921 | 0.6802 | 0.6648 | 0.6572 | 0.6518 | 0.6523 | 0.6498 |
| | | R@K | 0.064 | 0.118 | 0.1824 | 0.2451 | 0.2941 | 0.358 | 0.4102 | 0.4476 | 0.4926 | 0.5369 |
| | | F1@K | 0.1203 | 0.2038 | 0.2906 | 0.3647 | 0.4122 | 0.4607 | 0.5056 | 0.5314 | 0.5602 | 0.5812 |

Table 4: Top-k Recommendation on iKala Dataset

Next, we used the iKala dataset to further verify the robustness of PIN. In both the iKala Dataset and the I&B Dataset, the overall performance of PIN was similar. $PIN_{text}$ with inductive learning consistently yielded better results, achieving a recall rate close to 1. Additionally, in our experimental setup using the inductive learning paradigm, the product category nodes in the testing phase include nodes that were not seen during training. This indicates that the model maintains high accuracy even when processing unseen product category nodes.

We used inductive learning to address the cold-start scenario, as it is well-suited for launching new product categories. Given its effective performance, we applied the inductive learning settings to further analyze the effectiveness of different features. As can be seen from Table 1 & 3, in both datasets, $PIN_{text}$ consistently outperforms $PIN_{multi}$, which in turn outperforms $PIN_{image}$. We attribute this to the fact that images lack the context to fully convey an influencer's expertise in a particular domain. Text, on the other hand, provides a more comprehensive understanding of an influencer's proficiency with a specific product category. Although previous research on brand-influencer pairings often prioritized visual style and emphasized image features (Arifianto et al., 2018; Gan et al., 2019; Elwood et al., 2021; Kim et al., 2023), our product-centric research shows that text features are more effective in revealing the connection between influencers and product categories, thereby diminishing the importance of images in this context.

It's important to note that these findings do not imply that influencers should avoid using images in their posts. Images can enhance user experience and increase engagement, but text appears to be more effective for matching product categories with influencers in our PIN framework.

In conclusion, the appealing performance of PIN, with roc_auc consistently exceeding 0.95 across diverse datasets, validates the effectiveness of a product-centric approach in influencer marketing. Moreover, the framework's overall F1 score averages around 0.7 for influencer recommendation, not only exhibits a significant improvement over previous work, but also demonstrates the efficacy of using heterogeneous network structures and text features for modeling influencer marketing relationships and identifying product-influencer connections.

## 5 CONCLUSION AND FUTURE WORK

We proposed PIN, a product-centric framework for influencer recommendation that utilizes heterogeneous information networks (HIN) to expand the pool of potential influencers and uncover non-obvious collaborations for new products. Our experimental results using two different datasets confirm that the network structures can effectively model the complex relationships inherent in between influencers, product categories, and brands. In addition, we showed that text features are more crucial than images for identifying cooperation relationship between product categories and influencers. This helps reduce resources that are need to build the framework in real work applications. In short, this research contributes a novel and effective solution for influencer marketing, enabling businesses to discover new partnership opportunities and maximize the impact of their campaigns.

Future research may enhance product category prediction using supervised models trained on manually labeled data or leveraging unlabeled data from Instagram's shopping feature. Linking product categories to e-commerce descriptions could further improve text embeddings. Additionally, as brand profiles become richer, incorporating brand influence analysis could offer a more comprehensive understanding of influencer marketing dynamics. Further exploration into diverse brand types and their interaction with different influencer tiers could pave the way for more sophisticated machine learning applications in influencer marketing.

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

## A APPENDIX

The Appendix presents some prediction results using the PIN$_{\text{text}}$ model with inductive learning paradigm on iKala dataset. The study aims to demonstrate the model's ability to predict the suitability between a given product-category and an influencer, rather than illustrating the end-to-end prediction results of the entire system. First, we illustrated examples of correct predictions. Figure 6 displays a product from the category **Health & Beauty** > **Personal Care** > **Cosmetics** > **Skin Care** > **Lip Balms & Treatments** > **Lip Balm**s on Google Shopping. Figure 7 shows two accounts predicted by PIN as suitable for promoting this product category. The match is evident by the influencers' main pages featuring extensive beauty-related content, including lip balms and lipsticks. The ground truth also confirms that the two influencers have indeed promoted this product category.

As mentioned earlier, if a new product does not fit into the existing taxonomy categories, it can be assigned a shorter category path to be included in the network. To illustrate, we present two product categories with incomplete taxonomy paths. The first is **Animals & Pet Supplies** > **Pet Supplies** > **Cat Supplies**, encompassing cat-related products such as those shown in Figure 8. This category can be further divided into subcategories like Cat Litters, Cat Foods, and Cat Toys. The influencers predicted to be suitable for endorsing this category are shown in Figure 9. Although @stupidbank's main page in Figure 9 primarily features dog-related content, by clicking through their pages, we found that they also own a cat and have indeed promoted cat food. Thus, recommending them for promoting cat-related products is acceptable.

While the previous examples demonstrate the effectiveness of our model, we also want to highlight instances where our model made incorrect predictions. We again use cat-related products as example. Although our framework successfully predicted influencers for the broader **Cat Supplies** category, an incorrect prediction was made within the more specific category of **Animals & Pet**

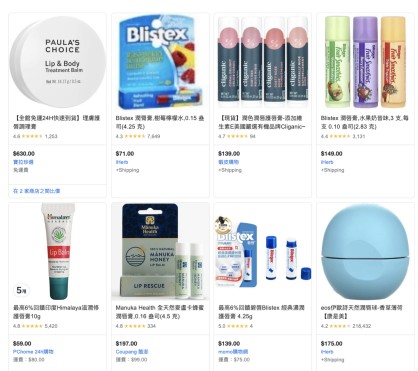

Figure 6: Products of the lip-balm category on Google Shopping

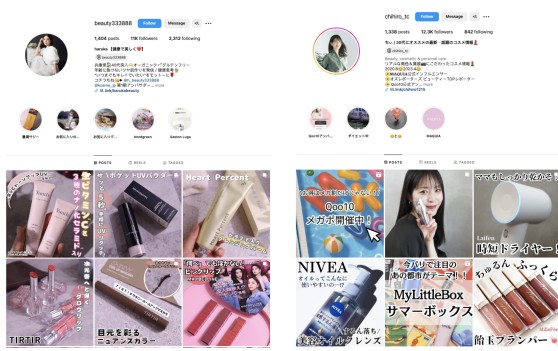

Figure 7: Influencers' Instagrams for the lip-balm product category

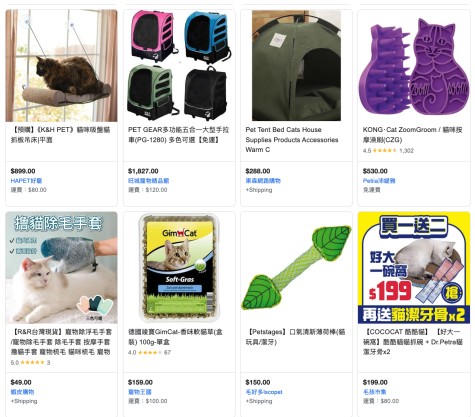

Figure 8: Products of the cat-supplies category on Google Shopping

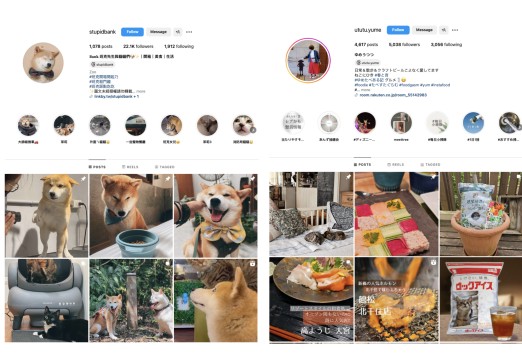

Figure 9: Influencers' Instagrams for the cat-supplies product category

**Supplies** > **Pet Supplies** > **Cat Supplies** > **Cat Food**. The influencer @doudou1109 in Figure 10, predicted to be suitable for promoting cat food, is actually a dog influencer and should not be paired with this product category. This mismatch may be attributed to our overreliance on text information alone. Since both cats and dogs are categorized as pets and animals, their text embeddings may be too similar to effectively differentiate them. This highlights a potential area for improvement in our model, such as incorporating additional data modalities or refining our text embedding techniques.

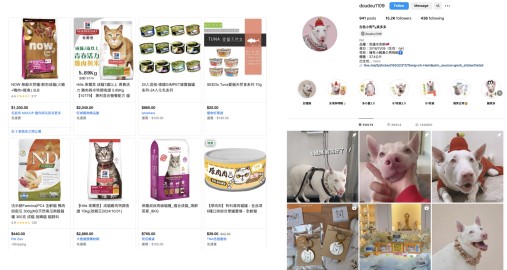

Figure 10: Products of the Cat Food category on Google Shopping and its predicted influencer

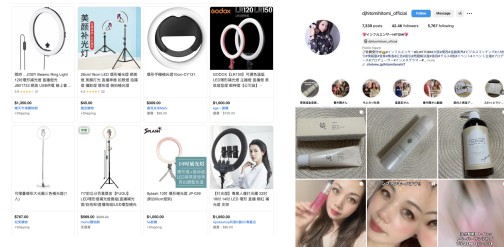

Figure 11: Products of the Studio Stand & Mount Accessories category on Google Shopping and its predicted influencer

Finally, we present another incorrect prediction, where our model predicted a collaboration that does not exist. Figure 11 shows some products in category **Cameras & Optics** > **Photography** > **Lighting & Studio** > **Studio Stand & Mount Accessories**. Common products in this category include photography stands and lighting equipment used for makeup tutorials or live streaming. Figure 11 shows an influencer who has never promoted products in this category, yet our model predicted a potential collaboration. While this is technically an incorrect prediction, it highlights the potential for PIN to expand the candidate pool of influencers. In this case, a beauty influencer could plausibly promote lighting equipment designed for makeup application, illustrating the model's ability to identify less obvious, but still relevant, influencer-product pairings.

The examples above demonstrate that PIN can effectively identify suitable influencers across various product categories. Although some predictions may be marked as incorrect due to the lack of prior collaboration, a closer manual inspection, such as in Figure 11, reveals that the predicted influencers are indeed appropriate for promoting the given product category. This highlights PIN's capability to match influencers to product categories based on their text content.

