# OpenReview forum: "ProdInfluencerNet: A Novel Product-Centric Influencer Recommendation Framework Based on Heterogeneous Networks"
_ICLR.cc/2025/Conference — Submitted to ICLR 2025_

### Official Review · Reviewer_WqaC · 2024-11-03

**Soundness:** 2
**Presentation:** 2
**Contribution:** 2
**Rating:** 3
**Confidence:** 4

**Summary:**

In this paper, the authors proposed ProdInfluencerNet (PIN), a product-centric framework designed for influencer recommendation. PIN models the complex relationships between brands, products, and influencers using Heterogeneous Information Networks. PIN can effectively recommend influencers for brands, even in new markets or with innovative products. The effectiveness of PIN is validated using real-world datasets collected from Instagram.

**Strengths:**

1.	The paper proposes an innovative, product-centric influencer recommendation framework, ProdInfluencerNet (PIN), which leverages Heterogeneous Information Networks (HINs) for effectively identifying influencers based on product categories, making it more robust for new or unique products.
2.	The use of a hierarchical Google Taxonomy for categorizing products demonstrates an impressive application of structured data to influencer marketing, enhancing the accuracy of influencer-product alignment.
3.	The inductive learning approach and extensive experiments on real-world Instagram datasets effectively showcase the model's ability to address cold-start issues and provide comparative results with other established frameworks.

**Weaknesses:**

1.	The network design of the heterogeneous graph is overly simplistic and lacks innovative adaptations tailored to the specific task.
2.	The equations are not adequately standardized, and relevant symbols lack proper definitions.
3.	The font size in Figure 3 is relatively small, and the colors are light, which may hinder readability for the audience.

**Questions:**

1.	In the era of short videos, I believe that finding suitable influencers for product promotion is a highly meaningful research direction. However, why did the authors not design a model architecture more tailored to this task, instead opting to use a standard heterogeneous graph?
2.	Expressions like "edge feat product" appearing directly in equations are highly unconventional. Why didn’t the authors adhere to standard mathematical notation for the formulas?
3.	Why is the final loss function not provided in the methodology section? This omission makes it difficult to understand how the model is optimized.

---

### Official Review · Reviewer_ijQt · 2024-11-03

**Soundness:** 2
**Presentation:** 3
**Contribution:** 2
**Rating:** 5
**Confidence:** 4

**Summary:**

This paper addresses the influencer recommendation problem in social media. To effectively tackle the cold-start challenge, the paper introduces ProdInfluencerNet (PIN), a Heterogeneous Information Network designed to better model the complex relationships between brands, products, and influencers. The proposed approach is validated using two Instagram datasets.

**Strengths:**

+ Problem: The influencer recommendation problem is both interesting and impactful. It aligns well with the Internet celebrity economy and should be a major recommendation topic for social media platforms.

+ Results: The figures are generally clear and informative. The reviewer appreciates the examples provided in Fig. 5, which help readers understand the content better.

**Weaknesses:**

- Baselines: There is only one baseline, GNN-IR (Park et al., 2024), used in the experiments. Additionally, the information for GNN-IR in Tables 1 and 2 appears to be incorrect. It is not very convincing to have only one straightforward GNN baseline, even though several configuration-sweeping experiments of PIN are conducted. An intuitive baseline might involve using products directly rather than product categories. It would also be beneficial to consider non-GNN baselines.

- Problem formulation and related work: The influencer recommendation problem seems to be similar to the regular recommendation problem. If this is the case, more methods like collaborative filtering or more advanced deep learning methods should be considered. If not, the unique challenges of influencer recommendation should be adequately discussed.

- Contributions: The primary contribution of this paper over the baseline GNN-IR is the use of a Heterogeneous Information Network, which is already widely used (e.g., in movie recommendation applications as shown in Fig. 2). While it is beneficial to use product categories to connect both sides, the technical contributions appear to be limited.

**Questions:**

1. Although there are some experiments regarding influencer feature modality (text, image, or multimodal), why is this feature modality not considered for brands and products?

2. For the two learning paradigms, is it possible to combine them for training and evaluation? The reason for this question is that it might be more practical to consider the overall performance of both paradigms in real recommendation scenarios.

---

### Official Review · Reviewer_cNtE · 2024-11-05

**Soundness:** 2
**Presentation:** 2
**Contribution:** 1
**Rating:** 1
**Confidence:** 5

**Summary:**

The paper addresses the influencer recommendation by modeling it as a Heterogeneous Information Network (HIN) where the product category is one of the core nodes beyond the influencer nodes and brand nodes. The product category is inferred from the product image and mapped into Google’s product taxonomy. The proposed model utilizes the GraphSAGE method. Experiments on two Instagram datasets are conducted to show the effectiveness.

**Strengths:**

S1: To extract more information from the product image seems promising.

**Weaknesses:**

W1: The contributions seem not significant. On lines 121-123, it says, “While GraphSAGE can theoretically handle unseen nodes, its real-world application remains untested. Our PIN experiment aims to address this gap and use GNN-IR as the benchmark.” So, what is the real technical contribution of this submission?

W2: The datasets are not large. On the filtered Influencer and Brand (I&B) Dataset, there are 3K influencers and 14K brands.  On the iKala Dataset, there are 3K influencer nodes and 15K brand nodes.

W3: The evaluation seems weak. There is only one baseline, GNN-IR. Why?

**Questions:**

Comments:

The following C1 an C2 are two relevant questions.

C1: The authors use the category information of the product. How about exploiting the product’s title/name information? Or post caption, e.g., “perfume A”, “toner C”, “blush B”, in section 3.2.4.

C2: The authors extract the classification labels (taxonomy) from the product image. How about extracting objects and attributes from the product image? In section 3.2.4 and Figure 5, we can have coarse categories of the products by recognizing the objects from the images, e.g., Perfume, Lipstick. What are the core contributions of using Google’s product taxonomy?

C3: On lines 214-215, it says “In such cases, lower-level taxonomy classes can be combined with product descriptions to form new product category nodes”. Do the experiments show such “new product” results? How do the "product descriptions" form product category nodes?

C4: On Lines 155-157, why the subscripts of B, K, and P are all the same m?

C5: the figure 3 is too small. When you print it out on the paper, the contents are difficult to read.

---

### Official Review · Reviewer_CkZs · 2024-11-08

**Soundness:** 3
**Presentation:** 3
**Contribution:** 2
**Rating:** 5
**Confidence:** 4

**Summary:**

The paper introduces PIN, a framework designed to recommend influencers for brands based on a product-centric approach. PIN utilizes HINs to effectively model the intricate relationships between brands, products, and influencers. A standout feature of PIN is its use of Google Taxonomy for categorizing sponsored post images through image classification techniques, which allows for accurate influencer recommendations even in new markets or with novel products. The framework employs an inductive learning approach, enhancing its capability to generalize and perform well with unseen product categories, thus addressing the cold-start issue effectively.

**Strengths:**

S1. The motivation of this paper is strong, which combines heterogeneous information networks with inductive learning to address the challenge of influencer recommendation for brands in new markets or with new products.

S2. The overall structure of the paper is clear and easy to follow.

S3. The paper seems can be easily adopted in industrial recommender systems. The paper provides an in-depth analysis of the role of text and image features in influencer recommendation, which is a guiding discovery for practical applications.

**Weaknesses:**

W1. The PIN framework relies on Google Taxonomy for product categorization, which may limit the framework's applicability in scenarios with other classification systems or custom categorization needs.

W2. Hinformation networks and deep learning models typically require substantial computational resources, which could affect its deployment in environments with limited resources. The authors should discuss this part in details.

W3. One target of this paper is to enhance the modeling of new items. However, there is compard with only one baseline GNN-IR, which heavily limits the technical contribution. For example, there are many recent HIN recommendation baselines and item cold-start recommendation models should be compared. This part is very improtant to evaluate the effectiveness of the model.

**Questions:**

See W1-W3. Further one question is listed below:

Q1. Although the PIN framework is effective, the paper does not discuss the interpretability of the model in detail, i.e., how to explain the choice of recommended influencers to users.

---

### Meta-Review · Area_Chair_hQxF · 2024-12-16

**Metareview:**

Four reviewers have provided detailed comments on this paper, and their scores are 5 (marginally below the acceptance threshold), 1 (strong reject), 3 (reject). The reviewers have concerns on the limited technique novelty of the work, the insufficient comparison with other baselines, and the weak evaluations. Some technique details on the method should be also clarified. As the authors have not provided rebuttal to the comments, there is no discussion on the review of the paper. Based on the reviewers' comments, I recommend to reject the paper.

**Additional Comments On Reviewer Discussion:**

The authors do not provide rebuttal.

---

### Decision · Program_Chairs · 2025-01-22

Reject